

**Dominant synoptic patterns associated with the decay process of**
**PM2.5 pollution episodes around Beijing**
Xiaoyan Wang[1,2,3*], Renhe Zhang[1,2], Yanke Tan[1,2], Wei Yu[1,4]
[1]Department of Atmospheric and Oceanic Sciences & Institute of Atmospheric Sciences, Fudan
University, Shanghai, China
[2]Big Data Institute for Carbon Emission and Environmental Pollution, Fudan University, Shanghai,
China
[3]Shanghai Institute of Pollution Control and Ecological Security, Shanghai, China
[4]Shanghai Ecological Forecasting and Remote Sensing Center, Shanghai, China
Correspondence to: Xiaoyan Wang (wangxyfd@fudan.edu.cn)


## Abstract

The variation in the concentrations of ambient PM2.5 (particles with an aerodynamic diameter less than 2.5 µm) generally forms a continuous sawtooth cycle with a recurring smooth increase followed by a sharp decrease. The abrupt decay of pollution episode is mostly meteorological in origin, and is controlled by the passage of synoptic systems. One affordable and effective measure for the quickly reducing PM2.5 concentrations in northern China is to wait for strong wind to arrive. However, it is still unclear how strong the wind needs to be and exactly what kind of synoptic system most effectively results in the rapid decay of air pollution episodes. PM2.5 variations over the 28 pollution channel cities of Beijing are investigated to determine the mechanisms by which synoptic patterns affect the decay processes of pollution episodes. This work shows more obvious day-to-day variations in PM2.5 concentration in winter than in summer, which implies that wintertime PM2.5 variations are more sensitive to meteorological factors. There were 365 decay processes from January 2014 to March 2020, and 97 of them were related to the effective wet deposition. 26%~43% of PM2.5 pollutant is removed by the wet deposition in different seasons. Two dominant circulation patterns are identified in summer, and the same three circulation types (CTs) are identified in the other three seasons based on the dry-day cases. The circulation patterns beneficial to the decay processes all exhibit a higher than normal surface wind speed, a negative relative humidity anomaly and positive divergence in the PM2.5 horizontal flux. In addition, CT1 in spring, autumn and winter is controlled by northeasterly wind and features the most significant horizontal net-outflow of air pollutants and effective upward spread of air pollutants to the free atmosphere, which promotes the abrupt reduction of local PM2.5 concentrations. CT2 is the most frequent synoptic pattern leading to decay processes in autumn and winter, and the domain region is located to the east of an anticyclone system. CT2 features a strong northwesterly wind of 2.98~3.88 m/s, the lowest relative humidity and the highest boundary layer height (BLH) among the three CTs, all of which are favorable for the reduction of PM2.5 concentrations. In CT3, a prevailing westerly wind anomaly occurs in the domain, with remarkable zonal divergence in the PM2.5 flux and strong horizontal wind shear in the near-surface under the boundary layer. PM2.5 concentrations show significant decreases of more than 37%, 41% and 27% after the passage of CT1, CT2 and CT3, respectively. A dry air mass with a positive BLH anomaly and the effective horizontal outflow of air pollutants are the main reasons for the abrupt decay phase in summer. PM2.5 concentrations after the decay process show a significant decreasing trend from 2014 to 2020, reflecting successful emission mitigation. Emission reductions have led to a 4.3~5.7 µg/(m$^3$.yr) decrease in PM2.5 concentrations in the 28 pollution channel cities of Beijing.



## 1. Introduction


PM2.5 pollution (particles with an aerodynamic diameter less than 2.5 μm) has become a severe
threat and challenge in China, especially in the Beijing-Tianjin-Hebei (BTH) region, and has
attracted significant concern regarding how to improve regional air quality (Che et al., 2019;Wang
et al., 2019a;Xia et al., 2016;Zhang et al., 2018;Mu and Zhang, 2014;Cai et al., 2017;Wang et al.,
2015). To avoid the severe negative impacts of air pollution on public health, the Chinese
government has issued a number of policies to improve the atmospheric environment (Ding et al.,
2019;Chen and Wang, 2015;Zhao et al., 2019;Li et al., 2018b). For example, in September 2013,
the State Council issued the Air Pollution Prevention and Control Action Plan (referred to as Clean
Air Action), which required the BTH region to reduce its PM2.5 concentrations by 25% within 5
years (China's State Council, 2013). With the deep research on the prevention and control of air
pollution, the regional effects of air pollution from cities in the pollution transmission channel in
the BTH region have been highlighted (China Daily, 2017). Therefore, the Work Plan for Air
Pollution Prevention and Control in Beijing, Tianjin, and Hebei and Surrounding Areas was released
in March 2017 (China's State Council, 2018). Much stricter, more comprehensive, and more
detailed prevention and control measurements were taken in the "2+26" cities, including Beijing,
Tianjin, and 26 other cities in the provinces of Hebei, Shandong, Henan and Shanxi. Due to the
persistent efforts towards emission mitigation, the air quality has shown significant improvement in
these 28 pollution channel cities in recent years (Zhang et al., 2019a;Zhang et al., 2019b;Zheng et
al., 2018;Wang et al., 2019d;Gui et al., 2020).
Meteorological conditions are considered as one of the important factors for the variation in ambient
PM2.5 pollution, especially for the temporal evolution of each air pollution episode (Zhang et al.,
2014;Ma and Zhang, 2020;Wang et al., 2019c). Even under the conditions of a significant decrease
in air pollutant emissions, similar to the COVID-19 lockdown period, PM2.5 pollution events still
occur frequently in the 28 pollution channel cities due to the unfavorable meteorological
background (Shi and Brasseur, 2020;Le et al., 2020;Huang et al., 2020b;Wang et al., 2020b; Wang
and Zhang, 2020b). Many studies have been conducted and have suggested that multiple
meteorological factors influence the emission of primary pollutants, the formation of secondary
particles and the processes of transport, accumulation and deposition of particles (Zhao et al.,
2020a;Huang et al., 2020c;Chen et al., 2019;Gong and Liao, 2019). High temperatures result in
greater emissions of PM2.5 precursors and secondary pollutants, and promotes photochemical
reactions, causing an increase in local PM2.5 concentrations (Zhang, 2017;Zhao et al., 2018b;Chen
et al., 2020). Humidity strongly affects PM2.5 concentrations in China, especially during severe
pollution episodes (Zhao et al., 2018a;Li et al., 2018a;Huang et al., 2020a). Higher humidity is





beneficial for the hygroscopic increase in aerosols and facilitates the formation of secondary
particles (Wang et al., 2019b;Zhao et al., 2017;Cheng et al., 2015;Xin et al., 2016). The cross-
regional transport and horizontal diffusion of pollutants are strongly determined by the wind field.
Southerly winds bring higher concentrations of air pollutants and more moisture, which enhances
the local air pollution in Beijing and the surrounding regions (He et al., 2020;Zhao et al., 2020b). In
addition to individual meteorological variables, synoptic circulation characteristics control the
formation and development of air pollution events (Wang et al., 2020a;Miao et al., 2020;Wang and
Zhang, 2020a; Liu et al., 2019). Monsoonal flows and cold frontal passages are the dominant
meteorological modes controlling the day-to-day variations in PM2.5 concentrations in the northern
China (Li et al., 2016;Wu et al., 2017;Zhang et al., 1996;Leung et al., 2018). Weak synoptic patterns
with high-pressure or persistent low-pressure systems favor the accumulation of pollutants, while,
strong synoptic patterns with large pressure gradients encourage the diffusion of pollutants (Cai et
al., 2020;Zhang et al., 2017;Zhang et al., 2020;Li et al., 2019). Severe haze events in the BTH region
are always accompanied by stagnant air conditions, stable stratification, weak surface wind, low
boundary layer height (BLH), and high relative humidity (Ma et al., 2020;Bi et al., 2014;Wang et
al., 2020c;Tang et al., 2016;Quan et al., 2020;Pei et al., 2020;Guo et al., 2019).
Most of the aforementioned studies focused on the synoptic pattern characteristics favorable for the
initiation and development of air pollution episodes in the BTH region. During the developing phase
of each PM2.5 pollution episode, the comprehensive effects of secondary aerosol formation,
hygroscopic increase and accumulation of particles lead to an increase in local PM2.5
concentrations, which usually takes several days from a clean situation to the outbreak of a heavy
haze (Sun et al., 2014;Wang et al., 2016;Pei et al., 2018). Both atmospheric chemistry and physics
processes play important roles in the developing phase of air pollution events (Gu et al., 2020;Yao
et al., 2018;Wang et al., 2018;Wang et al., 2010;Li et al., 2017;Gao et al., 2017). However, compared
to the developing phase, which typically features a smooth increase in air pollutant concentrations,
the decay phase of each pollution episode shows a sharp decrease in PM2.5 concentrations, often in
a few hours. The abrupt decrease in PM2.5 concentrations is purely meteorological in origin and is
controlled by the passage of synoptic systems, especially cold fronts, which terminate a severe air
pollution episode in the BTH region by strong winds (Zhu et al., 2016;Jia et al., 2008;Ji et al.,
2012;Xin et al., 2012). However, it is still unclear how strong the wind needs to be, exactly what
kind of synoptic systems can effectively terminate air pollution episodes in the BTH region, and
what mechanism is responsible for the rapid reduction in PM2.5 concentrations in a few hours. The
clarification of these issues will contribute to improving local air quality predictions. The variation
in air quality is generally consistent in the 28 pollution channel cities, especially in the decay phase
of pollution episodes, which indicates that the same synoptic system usually affects the whole region.





This study will focus on the region covering these 28 pollution channel cities and reveal the synoptic
circulation pattern that dominates the decay process of PM2.5 pollution events.

## 2. Data and Method

### 2.1 Dataset

The daily mean observed PM2.5 concentrations in the 28 pollution channel cities from January 2014
to March 2020 were obtained from the Ministry of Ecology and Environment of the People's
Republic of China (https://www.aqistudy.cn/historydata/). Fig. 1 shows the location of the 28
pollution channel cities and their annual mean PM2.5 concentrations from 2014 to 2019. The four-
times-daily dataset of the fifth-generation European Centre for Medium-Range Weather Forecasts
(ECMWF ERA5) atmospheric reanalysis dataset with a resolution of 0.5°
(https://cds.climate.copernicus.eu/cdsapp#!/dataset/10.24381/cds.bd0915c6?tab=form) was used to
describe the meteorological characteristics and synoptic circulation classification.
The divergence of local PM2.5 flux can be taken as a metric for the PM2.5 budget in the specific
region, with positive divergence indicating net outflow of air pollutants from the domain region,
and vice versa. The daily mean divergence of the PM2.5 flux over the region of 34-40° N and 112-
118° E is calculated according to Eq.(1):
$$D = D_Z + D_m = \frac{\partial}{\partial x}(UQ) + \frac{\partial}{\partial y}(VQ) = \sum_{i=1}^{n}\frac{(U_{Ei}Q_{Ei}-U_{Wi}Q_{Wi})}{2\Delta X} + \sum_{j=1}^{m}\frac{(V_{Nj}Q_{Nj}-V_{Sj}Q_{Sj})}{2\Delta Y} \quad (1)$$
where $D_z$ and $D_m$ are the zonal and meridional components of the net divergence of PM2.5 flux for
the specific region. The parameters $n$ and $m$ indicate the meridional and zonal grid numbers of the
domain. The subscripts $E$ and $W$ mark the variables at the longitudes of the eastern and western
boundaries of the domain. Similarly, the subscripts $S$ and $N$ represent the values at the latitudes of
the southern and northern boundaries. $U_{Ei}$ (units in m/s) indicates the 10 m zonal wind in the $ith$ grid
of the eastern boundary of the domain. $Q_{Nj}$ (units in μg/m³) is the spatially interpolated PM2.5
concentration in the $jth$ grid at the latitude of the northern boundary. $\Delta X$ and $\Delta Y$ represent the zonal
and meridional distance of each grid (units in meters). Due to the limited information on the vertical
distribution of PM2.5 and the horizontal winds are closely related with PM2.5 concentration as
revealed by previous studies, the horizontal divergence of PM2.5 flux is used to evaluate the net
inflow and outflow of local air pollutants in this study.

### 2.2 Thresholds for the decay process of air pollution episodes

Fig. 2 shows the daily PM2.5 concentration variations of the 28 pollution channel cities from



January to March 2019. PM2.5 concentrations exhibit a recurring smooth increase followed by a
sharp decrease, which is known as a sawtooth cycle (Jia et al., 2008). During the developing phase
of each pollution episode, the PM2.5 concentrations show the same smoothly increasing trend with
slight differences in the rate of increase and magnitude in the 28 pollution channel cities (i.e., an
average increase trend of $10.37 \pm 42.2\, \mu g/(m^3 \cdot day)$ during January to March 2019). The
inhomogeneity of the PM2.5 concentration increase in the 28 cities, indicating by the large standard
deviation of increase trends (approximate four times the magnitude of increase trend), may be due
to the complicated physiochemical processes of haze formation. By contrast, as shown by dotted
lines in Fig. 2, regional synchronous decreases in PM2.5 concentrations occur in the decay phase of
pollution episodes with an average trend of $-50.06 \pm 46.83\, \mu g/(m^3 \cdot day)$. Most of the
consistent improvements in air quality in the decay phase can be attributed to the effects of the
synoptic system. Therefore, in this study, if more than 40% of the 28 pollution channel cities with
the day-to-day PM2.5 concentrations decreased by 30% (relative to the value of the previous day)
or more than 60% of the channel cities with PM2.5 concentrations decreased by 30% in two
successive days, it can be defined as the occurrence of the decay phase of pollution episodes. If two
consecutive days were defined as the decay phase, only the first day was selected be vailed and
retained. In total, 365 days are identified as the decay phase of pollution episodes from January
2015 to March 2020 (see Fig. 4) and are used for the synoptic pattern classification.
**2.3 Method of synoptic circulation classification**
The T-mode principal component analysis (PCA) method was used to objectively classify the type
of synoptic system dominating the decay phase of pollution episodes, as this method has an
outstanding performance in terms of the reproduction of predefined types and temporal-spatial
stabilities (Huth et al., 2008;Cavazos, 2000;Tie et al., 2015;Valverde et al., 2015;Xu et al., 2016).
The T-mode PCA has been widely used to investigate the general circulation patterns, climate
change and air quality and has been incorporated into the European Cooperation in Science and
Technology (COST) plan 733 toolbox (COST733: http://cost733.geo.uni-augsburg.de/cost733wiki)
(Philipp et al., 2014). The daily mean geopotential height (Z), U and V components at 925 hPa on
the 365 decay phase days are used for synoptic pattern classification. To exclude the effects of
seasonal variation on atmospheric circulation and to ensure that different synoptic patterns in the
same season are comparable, the T-mode PCA method is applied to the four seasons respectively.
The target region is 32-44° N and 110-122° E, as shown in Fig. 1. For each season, the three input
data matrixes (U, V and Z) have temporal and spatial dimensions, with spatial grids and time series
represented by rows and columns, respectively. To speed up computations of the T-mode PCA in
the COST733 toolbox, each matrix is first divided into 10 subsets. Then, the principal components



(PCs) are determined using the singular value decomposition for each subset, and an oblique
rotation is applied to the PCs to achieve better classification effects. The 10 classifications based on
the subsets are evaluated by the chi-square test and the subset with the highest sum is selected and
assigned to a type.

## 3. Results

### 3.1 Identification of the occurrence of the decay process of air pollution episodes

The magnitude of the day-to-day variation in PM2.5 concentrations is an important metric for
recognizing the occurrence of the decay phase of air pollution. Fig. 3 shows the frequency of the
relative day-to-day PM2.5 concentration differences in the 28 pollution channel cities during the
period of January 2014 to March 2020. Table 1 summarizes the occurrence frequency of the day-
to-day PM2.5 differences in the specific segment. It shows that a fatter-tailed probability distribution
exists in winter than in summer; thus, winter features a lower probability of weak PM2.5 variations
and a higher probability of strong PM2.5 variations, indicating greater day-to-day variability in
PM2.5 concentrations. In winter, 8.6% of PM2.5 concentrations decreased by over 60%, and 14.9%
increased by more than 80%, whereas, in summer, the values were only 2.4% and 6.6%. A total of
38.3% of the cases show day-to-day PM2.5 variations within the range of -20% to 40% in winter,
but where a total of 55.6% is observed in summer. The PM2.5 variations in spring and autumn
exhibit almost the same distribution patterns, with a relatively higher frequency of strong PM2.5
variations in autumn. Generally, the probability distributions in spring and autumn are between
those of summer and winter. The stronger day-to-day decreases in PM2.5 concentrations,
particularly the sharp wintertime reductions, may be attributable to the passage of a cold front
synoptic system, and the results suggest that the winter PM2.5 variations are the most sensitive to
synoptic patterns.
According to the occurrence of day-to-day PM2.5 differences in the 28 pollution channel cities, i.e.,
thresholds for the decay phase of air pollution episodes in Section 2.2, 365 decay processes have
been recognized from January 2014 to March 2020. 97 of the 365 decay phases have effective
precipitation more than of 10 mm/day, in which case the abrupt decrease in ambient PM2.5
concentrations are assumed to be related to wet deposition. Only the decay processes on dry days
are involved in the synoptic pattern classification in the following work. Figure 4 shows the annual
cycle of the decay process frequencies in the specific year. In most years, the figure shows a two-
peak annual cycle of the decay phase frequency with a valley in summer, and the valley becomes
deeper after removing the rainy cases. There are 105 (105), 62 (21), 86 (56) and 112 (109) decay



process days in spring, summer, autumn and winter for all (dry-day) cases, respectively.
Approximately 70% of the regional sharp reduction in summer can be attributed to the effect of wet
deposition.
**3.2 Classification of the synoptic circulation dominating the decay processes of air pollution**
**episodes**
T-mode PCA circulation classification has been applied to the dry-day decay process in individual
seasons. Figs. 5 and 6 show the original and anomalous circulation patterns at 925 hPa under each
circulation type (CT) condition. Two dominant circulation types (CTs) are identified in summer, and
three CTs are identified in the other seasons. The three dominant CTs in spring show almost the
same pattern as those of autumn and winter, and only the occurrence frequency of the CTs differ
among the seasons. The strong prevailing northwesterly wind in the CT2 condition is the commonly
accepted synoptic circulation favorable for the rapid decay of pollution episodes in the BTH region,
and CT2 is also the most frequent CT for the decay phase in autumn and winter. A large-scale high-
pressure system covers the region of central-western Mongolia, northern Xinjiang, Inner Mongolia
and Shaanxi Province in China. Deep low pressure is situated in the northeastern China and northern
Japan. The BTH region is located between the east of the anticyclone and west of the cyclone, and
is dominated by strong northwesterly surface winds with the speeds of 2.98~3.88 m/s in different
seasons. The northwesterly wind corresponds to the significant northerly wind anomaly, which is
beneficial for the transport of cold, clean and dry air masses southward. Although it shows
downward motion due to the upper westerly wind passing the leeward side (see Fig. 7), the other
meteorological variables summarized in Fig. 8 reveal that the highest wind speed, the highest
boundary layer height (BLH) and the lowest relative humidity occur under CT2 conditions, all of
which are favorable for the reduction of PM2.5 concentrations. Fig. 9 exhibits the distribution of
PM2.5 flux divergence over the region of 34-40° N and 112-118° E, and its zonal and meridional
components, with positive divergence indicating net horizontal outflow of air pollutants from the
BTH region, and negative divergence indicating the opposite. The PM2.5 flux divergence is found
to have positive values in all three CTs, indicating that the local ambient PM2.5 concentrations
decrease with the removal of the polluted air mass or the replacement with clean air. The positive
divergence of the PM2.5 flux in CT2 is mainly contributed by the meridional outflow, which
highlights the effects of the northerly wind anomaly. Clean, dry and strong northwesterly winds in
the CT2 condition are the major drivers of the decay process of air pollution episodes.
In CT1 in spring, autumn and winter, a surface high-pressure system initiates from the Siberian
region and slants forward to central Inner Mongolia and the BTH region, resulting in a position that
is more northeastward than the anticyclonic circulation in CT2. Most areas in China are controlled



248 by a high-pressure system. The BTH region is located within on southeastern edge of the high-

249 pressure center with an anticyclonic horizontal wind shear in the domain. The average surface wind

250 speed is of 2.63~3.02 m/s, which is higher than the seasonal mean but not as high as that under CT2

251 conditions. Although all the surface wind speed, BLH and relative humidity show favorable patterns

252 for air pollutant diffusion under CT1 conditions, the magnitudes of the above anomalies are not as

253 high as those under CT2 conditions. Therefore, there must be other mechanisms responsible for the

254 decay process of pollution episode that are distinct from those of CT2, as is generally believed.

255 According to the anomaly pattern in Fig. 6, the BTH region is located at the south of the anticyclone,

256 which is dominated by a remarkable northeasterly wind anomaly. This northeasterly wind anomaly

257 brings clean and dry air masses to the BTH region, and increases the outward and southward

258 transport of local air pollutants in the meanwhile, which results in the negative relative humidity

259 anomaly shown in Fig. 8. The net divergence of air pollutants (i.e., positive divergence of the PM2.5

260 flux in Fig. 9) is the most significant under CT1 conditions, indicating the contribution of horizontal

261 transport to the rapid decay of pollution episodes. In terms of vertical anomaly circulation, the BTH

262 region is located under the east of a high-level ridge and west of a high-level trough (figure not

263 shown here), where there is often upper level convergence and cause the surface high-pressure

264 anomaly to get higher (see Fig. 6). The upper level convergence leads to the vertical sinking in the

265 east of the BTH region, which also delivers upper dry and clean air to the surface. In addition, as

266 shown in Fig. 7, the significant clean vertical sinking airflow in the east of the BTH region combined

267 with the surface easterly wind anomaly results in air movement westward across the domain and

268 climbs up along the western mountain region. The upward flow carries the near-surface air

269 pollutants to the upper level of the boundary layer, where the pollution quickly spreads to the free

270 atmosphere due to the effective entrainment caused by the strong wind shear at the top of the

271 boundary layer (see Fig. 7). In general, the remarkable horizontal net-outflow of air pollutants,

272 negative humidity anomaly and effective outward spread of air pollutants to the free atmosphere

273 promote the abrupt reduction of local PM2.5 concentrations.

274 CT3 is the dominant synoptic pattern for the decay process in spring, with the highest frequency of

275 47%, compared with frequencies of 30% and 17% in autumn and winter. In this kind of circulation

276 pattern, there is only a closed low-pressure system located over the northeastern China, with large

277 pressure gradients around the cyclone and weak gradients over most parts of China. The BTH region

278 borders the cyclone system to the northeast, which leads to a prevailing westerly wind with speeds

279 of 2.29~3.07 m/s. The low-pressure and westerly wind features are more significant based on the

280 anomalous circulation at 925 hPa in Fig. 6, especially in winter. In the upper 500 hPa, a deep trough

281 persists in the northern BTH region, bringing cold air masses from the northwest. Similar to CT1

282 and CT2, negative relative humidity anomalies and positive surface wind speed anomalies are



favorable for the decay of pollution episodes. Given the distribution of the BLH, there is no significant positive anomaly signal in CT3, unlike in CT1 and CT2. Although a moderate BLH is observed under CT3 conditions, strong vertical wind shear occurs near the surface, as shown in Fig. 7, which improves the more uniform vertical distribution of air pollutants in the boundary layer. Moreover, obvious horizontal PM2.5 divergence also provides a possibility for the decay of air pollution episodes. To be more precise, the zonal divergence of the PM2.5 flux that dominates the net divergence of the whole region, rather than the meridional component as the other two circulation patterns. The inflow of clean and dry air masses combined with the good performance of boundary layer mixing are the main reasons for the immediate improvement of air quality when CT3 occurs.

In terms of the synoptic patterns in summer, two CTs are classified excluding the effects of wet deposition. According to the circulation anomaly in Fig. 6, the synoptic pattern of CT1 in summer is similar to that of CT3 at 925 hPa in other seasons, which is dominated by a northeastern cyclonic circulation. Dry northwesterly wind occurs in the BTH region, reducing the local relative humidity. As shown in Fig. 8, the BLH is higher than the seasonal average, indicating an increase in vertical diffusion space. The zonal positive divergence of the PM2.5 flux is offset by the negative value in the meridional direction. The effect of horizontal transport of air pollutants can be ignored in this situation. Therefore, the decay process of the air pollution episode in the CT1 condition can be attributed to the dry air mass and higher than normal BLH.

In the anomaly pattern of the CT2 condition in summer, the BTH region is located between the southern portion of a high-pressure system and the northern portion of a low-pressure system, and is affected by the prevailing northeasterly surface wind. Clean air masses are transported to the BTH region along with the northeasterly wind, which can be confirmed by the positive divergence in the PM2.5 flux in both zonal and meridional directions. Both the negative relative humidity and positive BLH anomalies in CT2 are beneficial for the reduction of surface PM2.5 concentrations, but the magnitude of the anomaly is not as high as those of the CT1 condition. There is no favorable signal for the diffusion of surface PM2.5 in terms of the vertical motion in the two synoptic patterns in summer. It is the effective horizontal outflow that promotes the decay process of pollution episodes.

**3.3 Synoptic circulation effects on the PM2.5 pollution**

Section 3.2 shows different physical mechanisms for the rapid decay of air pollution episodes in the region covering the 28 pollution channel cities. Fig. 10 exhibits the relative difference in PM2.5 concentrations between the day before and after the occurrence of the specific synoptic CTs. The average PM2.5 differences in the 28 pollution channel cities are summarized in Table 2. Unsurprisingly, it shows a remarkable decrease in PM2.5 concentrations when all the circulation



patterns dominate the decay process occurs, but it is worth noting that the magnitudes of the decline
vary according to the synoptic patterns. For the case of spring, autumn and winter, CT2 conditions
demonstrate the most significant effects on the abrupt reduction in PM2.5 concentrations, with a
more than 40% day-to-day decrease in PM2.5 concentrations in the 28 pollution channel cities in
all three seasons. CT1 conditions are second in terms of the circulation influence in the decay
process of PM2.5 pollution episodes. The PM2.5 concentrations decrease quickly by 37.2%, 40.1%
and 36.9% when CT1 conditions occur in spring, autumn and winter, respectively. The CT3
conditions, which are dominated by westerly winds, show a relatively weak ability on control the
decay process of PM2.5 pollution episodes. Air quality improves by approximately 26~29%
compared with the previous day due to the occurrence of CT3 conditions. In summer, PM2.5
concentrations decrease more significantly with the occurrence of CT1 conditions than with the
occurrence of CT2 conditions, indicating more effective diffusion under northwesterly winds than
under northeasterly airflow. Wet scavenging is an effective method for the rapid decay of air
pollution episodes, especially in wintertime. PM2.5 concentrations drop sharply after the occurrence
of precipitation, with decreases of more than 35% in spring, autumn and winter. 26.2% of PM2.5
pollution is removed by the wet deposition in summer, which is the lowest rate among the four
seasons. The relatively clean background may account for the weak wet deposition effects in
summer.
Fig. 2 shows the sawtooth cycle variation in PM2.5 concentrations with a smooth increase followed
by an abrupt decrease. However, the PM2.5 concentrations do not always increase gradually before
the decay of the pollution episode. Here, the sawtooth cycle is divided into developing and decay
phases, and the interval stage between two decay phases is defined as the developing phase of a
specific pollution episode. As shown in Fig. 11, when the duration of the developing phase is less
than 3-days, air pollutants accumulate gradually to a maximum until the occurrence of decay process
occurs. However, if the developing phase is longer than 3-days, the highest PM2.5 concentrations
occur on 1-3 days before the passage of a favorable synoptic system, which indicates that the
developing mature stage of pollution episodes (with high level concentrations) usually persist for
several days.
The duration of the developing phase not only changes the shape of the sawtooth cycle but also
affects the maximum PM2.5 concentrations during the pollution episode, as shown in Fig. 12. Most
of the durations of the developing phase are concentrated in the period of shorter than 5-days in
spring, autumn and winter, with average durations of 5.53, 5.86 and 5.36 days, respectively.
Typically, for the cases in spring and autumn, when the durations are less than 5 days, the maximum
PM2.5 concentrations during the specific air pollution episode increase with an increase in the
developing phase durations; but the concentrations remain unchanged if the duration longer than 5





days. In winter, the maximum PM2.5 concentrations in a specific sawtooth cycle continue to
increase with increases in the interval between two decay processes. Wintertime air pollution can
be exacerbated by the long-term absence of an effective decay process. The frequency of favorable
circulation patterns is relatively lower in summer, which leads to an effective decay process
occurring every 7.45 days. The maximum PM2.5 concentrations display an upward tendency with
increases in the developing stage durations, but there are some small fluctuations in the mean value
of the highest PM2.5 concentration due to the limited samples in summer.
Emission and meteorological elements are taken as the two most important factors controlling the
variation in PM2.5. Many efforts have been made to mitigate the air pollutant emissions in the 28
pollution channel cities, which have achieved remarkable improvements in air quality in recent
years. However, because obvious interannual difference of the meteorological conditions are
observed, there is uncertainty in the evaluation of emission reductions based on the observed PM2.5
concentrations. The quantitative evaluation of the effects of emission reduction measures on the
PM2.5 concentration variation has been a challenge for policy makers and stakeholders. Here, only
the PM2.5 concentrations observed on the days of decay processes are compared, which excludes
the different effects of meteorological conditions and evaluates the pure effects of emission
reduction from a certain perspective. Fig. 13 shows a significant decline in seasonal mean PM2.5
concentrations from 2014 to 2020 in the 28 pollution channel cities. This figure also shows almost
the same rates of decrease in all four seasons, with relatively smaller decreases of 4.8 and 4.3
μg/(m³.yr) in spring and winter and greater decreases of 5.7 and 5.2 μg/(m³.yr) in summer and
autumn, respectively. The slight difference in the seasonal decreasing tendency is possibly due to
difference in the main sources of air pollutant emissions.

## 4. Conclusions and Discussion

The variation in ambient air pollutant concentrations generally represents a continuous sawtooth
cycle with a recurring smooth increase followed by a sharp decrease. The combined effects of
emissions, secondary formation of particles and unfavorable meteorological conditions that trigger
the initiation and development of the specific PM2.5 pollution episodes over several days. In
contrast, the abrupt decay of pollution episodes is mostly due to the passage of favorable synoptic
patterns, and it usually takes a few hours for conditions to transition from hazy to clean air condition.
The detailed atmospheric circulation features and the mechanisms through which they affect the
decay processes of pollution episodes are discussed in this work. A relatively fatter-tailed probability
distribution of day-to-day PM2.5 concentration over the 28 pollution channel cities is observed in
winter compared with summer, indicating that winter features a lower probability of weak PM2.5



variations and a higher probability of strong PM2.5 variations. The probability distribution of day-
to-day PM2.5 variations in spring and autumn lies between those of summer and winter. A total of
365 decay processes were recognized from January 2014 to March 2020 based on the regional
variation in the day-to-day PM2.5 concentration difference. 97 of the 365 decay phases were related
to the effective wet deposition, and most of them occurred in summer. For the dry-day decay
processes, 105, 21, 56 and 109 cases occurred in spring, summer, autumn and winter, respectively.
The intervals between two continuous decay processes are 5.53, 7.45, 5.86 and 5.36 days from
spring to winter, respectively.
T-mode PCA circulation classification has been applied to the dry-day decay process of the specific
season. Two dominant circulation patterns are identified in summer; three circulation types for the
other three seasons, in addition, it shows almost the same circulation patterns in the three seasons.
The results show that, in all the circulation patterns, a higher than normal surface wind speed, a
negative relative humidity anomaly and positive divergence in the PM2.5 horizontal flux benefit the
decay processes. However, there are some distinctive features among the different CTs. In spring,
autumn and winter, the dominant CTs are controlled by northeasterly, northwesterly and westerly
surface wind anomalies, respectively. With the prevailing northeasterly flow in CT1, clean and dry
air masses are transported to the BTH region, resulting in the most significant positive divergence
of air pollutants. Moreover, in the vertical direction, air pollutants carried by ascending motion are
quickly mixed into the free atmosphere with the aid of strong vertical wind shear at the top of the
boundary layer. The combined effects of horizontal and vertical diffusion lead to a reduction in
PM2.5 concentrations by over 36%. The most frequent pattern for the decay phase in autumn and
winter is CT2, in which the BTH region is located in the eastern portion of an anticyclone system
with strong northwesterly wind. CT2 has the highest surface wind speed, lowest relative humidity
and highest BLH among the three CTs, all of which are favorable for the quick decay of pollution
episodes. The PM2.5 concentrations in the 28 pollution channel cities decrease by more than 40%
after CT2 occurs. In CT3, the BTH region borders the cyclone system to the northeast, which leads
to a prevailing westerly wind anomaly. In addition to the effective zonal divergence in the PM2.5
flux, strong horizontal wind shear in the near-surface improves the more uniform vertical
distribution of air pollutants in the boundary layer. After the passage of CT3, 26~29% of local air
pollutants are typically removed. The two dry-day circulation patterns in summer are similar to the
synoptic patterns of CT1 and CT3 in the other three seasons. A dry air mass with a positive BLH
anomaly and the effective horizontal outflow of air pollutants are the main reasons for the abrupt
decay phases in summer. The average PM2.5 concentrations on decay process days show a
significant decreasing trend from 2014 to 2020, which indicates the success of emission mitigation
efforts. Emission reductions have led to a 4.3~5.7 µg/(m$^3$.yr) decrease in PM2.5 concentrations in

10010

the 28 pollution channel cities.

**Code/Data availability:** Daily PM$_{2.5}$ concentration observations at the 28 channel cities were
obtained from the website of Ministry of Ecology and Environment of the People's Republic of
China (http://106.37.208.233:20035). Daily four times ECMWF ERA5 dataset during 2014 to 2020
are    downloaded    from    https://www.ecmwf.int/en/forecasts/datasets/reanalysis-datasets/era5.
Atmospheric circulation classification was conducted using European Cooperation in Science &
Technology (COST) plan 733 (cost733class software), which can be downloaded at
http://cost733.met.no.

**Author contributions:** XW and RZ designed research. XW, YT and WY performed the analyses
and wrote the paper. All authors contributed to the final version of the paper.

**Competing interests:** The authors declare that they have no conflict of interest.

**Acknowledgements:** We thank the support of Fudan University-Tibet University Joint Laboratory
For Biodiversity and Global Change. This research has been funded by the National Natural Science
Foundation of China (grant nos. 41790470, 41805117 and 41975075).

**Figure captions:**

Figure 1. Annual mean PM2.5 concentrations in the 28 pollution channel cities of Beijing from 2014
to 2019 (units: μg/m$^3$).

Figure 2. Time series of daily mean PM2.5 concentrations in the 28 pollution channel cities from
January to March 2019 (units: μg/m$^3$).



Figure 3. Probability distribution of the relative day-to-day difference of PM2.5 concentrations. The
relative difference is based on the PM2.5 concentration on the previous day. The distributions in
spring and autumn are combined in the upper panel, and cases in winter and summer are shown at
the bottom.

Figure 4. Monthly cumulative occurrence of the decay processes of pollution episodes. The orange
curve indicates the decay process occurrences on dry days. In total, 365 decay processes are
identified from January 2014 to March 2020, and 97 of them are associated with precipitation levels
greater than 10 mm/day.

Figure 5. Distribution of the geopotential height (shaded, units: $m^2/s^2$) and wind fields at 925 hPa
for each circulation type. The number over each subplot indicates the occurrence frequency of the
specific circulation type. The solid blue box is the location of the domain region covering the 28
pollution channel cities.

Figure 6. Distribution of the geopotential height anomalies (shaded, unit: $m^2/s^2$) and wind field
anomalies at 925 hPa for each circulation type.

Figure 7. Zonal averaged profile of the distribution of vertical wind shear anomalies in the domain
region (shaded, units: m/(s.100 m) and the vertical and zonal circulation anomalies. The green line
indicates the average location of the top of the boundary layer. Zonal wind shear, circulation and
boundary layer height are the average values between 34-40° N. The two dashed lines are the eastern
and western boundaries of the domain (112 to118° E). The grey region indicates the average altitude
between 34-40° N.

Figure 8. Boxplot of surface wind speed, boundary layer height (BLH), sea level pressure (slp) and
relative humidity (RH) for each circulation type. The dashed line indicates the seasonal mean of the
specific variables.

Figure 9. Boxplot of the divergence of PM2.5 flux over the region of 34-40° N and 112-118° E. The
daily divergence is calculated based on the Eq.(1). Zonal and meridional components are the first



and second terms of the formula.

Figure 10. Distribution of the daily mean PM2.5 concentrations before and after the occurrence of
decay processes of pollution episodes in the 28 pollution channel cities. The hollow box indicates
the concentration on the decay phase day, and the solid box is the value on the previous day. The
relative differences in the PM2.5 concentrations after the occurrence of decay process are
summarized in Table 2. The number at the top of each box indicates the sample size used for the
boxplot. The number in the first line is the sample size of the "before" case; and the second line is
for the "after" case.

Figure 11. The day of the maximum PM2.5 concentration during each pollution episode varies with
the duration of the developing phase.

Figure 12. The density plot of the maximum PM2.5 concentration according to the duration of the
developing phase of pollution episodes. Daily PM2.5 concentrations are normalized by their
monthly mean value to exclude the effects of seasonal and interannual variations in air quality. A
warmer color indicates a higher density of scatter. Pentagrams mark the average maximum PM2.5
concentration for the specific duration period.

Figure 13. Variations in the average PM2.5 concentration on all the decay phase days from 2014 to
2020. The black hollow circles indicate the mean PM2.5 concentration in each year. The black line
is the fitting line based on the mean value. The number in the subplot is the linear trend of the fitting
line.





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


Table 1. Frequency of the relative day-to-day PM2.5 difference within the specific range.

| Relative Difference (%) | <-80 | -80~-60 | -60~-40 | -40~-20 | -20~0 | 0~40 | 40~80 | 80~120 | >120 |
|---|---|---|---|---|---|---|---|---|---|
| MAM | 0.4 | 4.4 | 9.0 | 13.5 | 17.2 | 31.6 | 14.3 | 5.4 | 3.8 |
| JJA | 0.2 | 2.2 | 7.6 | 15.6 | 20.7 | 34.9 | 12.0 | 4.3 | 2.3 |
| SON | 1.3 | 5.2 | 9.1 | 12.2 | 14.8 | 29.4 | 15.2 | 6.7 | 5.7 |
| DJF | 1.9 | 6.7 | 9.7 | 12.5 | 13.1 | 25.2 | 15.3 | 7.9 | 7.2 |



Table 2. The average relative difference of PM2.5 concentrations before and after the occurrence of
decay processes (i.e., $(PM_t-PM_{t-1})/PM_{t-1}*100$, where $PM_t$ is the daily mean PM2.5 concentration on
the decay phase day).

| % | CT1 | CT2 | CT3 | Wet deposition |
|---|---|---|---|---|
| MAM | -37.2 | -44.8 | -28.2 | -40 |
| JJA | -34.5 | -20.4 | // | -26.2 |
| SON | -40.1 | -42.9 | -26.9 | -35.8 |
| DJF | -36.9 | -41 | -29.3 | -43.9 |





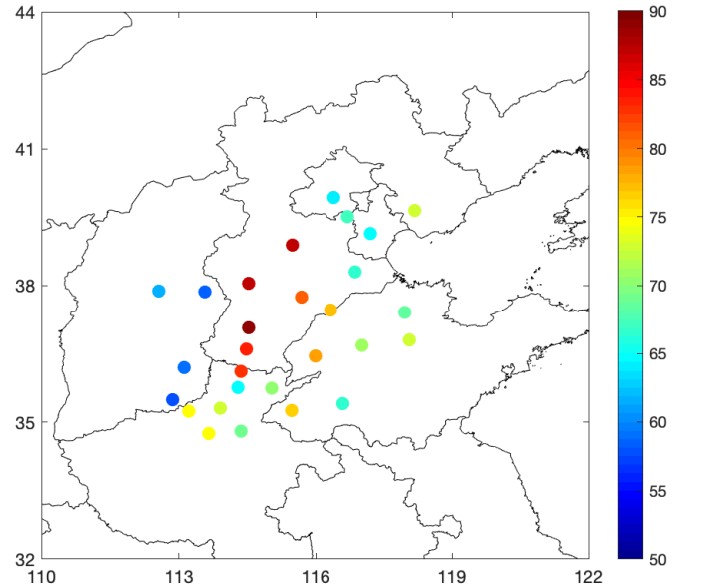


Figure 1. Annual mean PM2.5 concentrations in the 28 pollution channel cities of Beijing from 2014
to 2019 (units: μg/m$^3$).

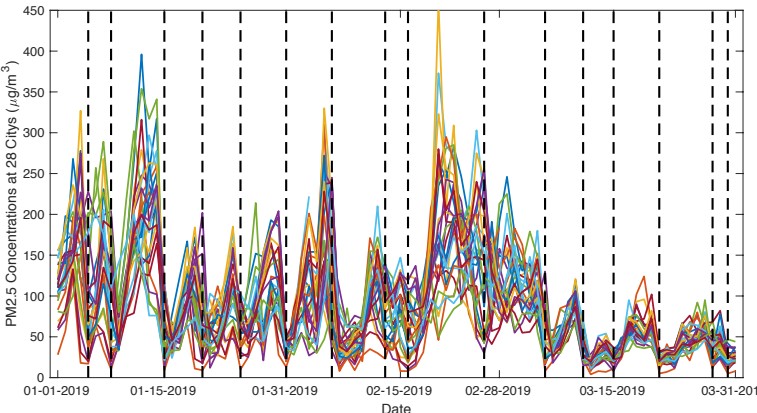


Figure 2. Time series of daily mean PM2.5 concentrations in the 28 pollution channel cities from
January to March 2019 (units: μg/m$^3$).






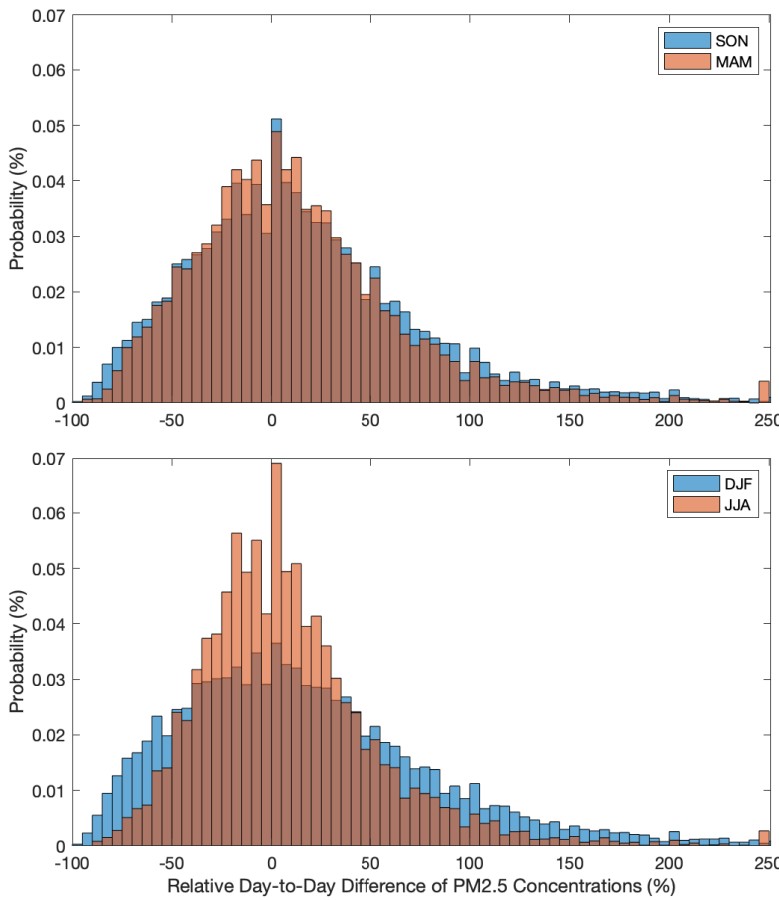


Figure 3. Probability distribution of the relative day-to-day difference of PM2.5 concentrations. The relative difference is based on the PM2.5 concentration on the previous day. The distributions in spring and autumn are combined in the upper panel, and cases in winter and summer are shown at the bottom.



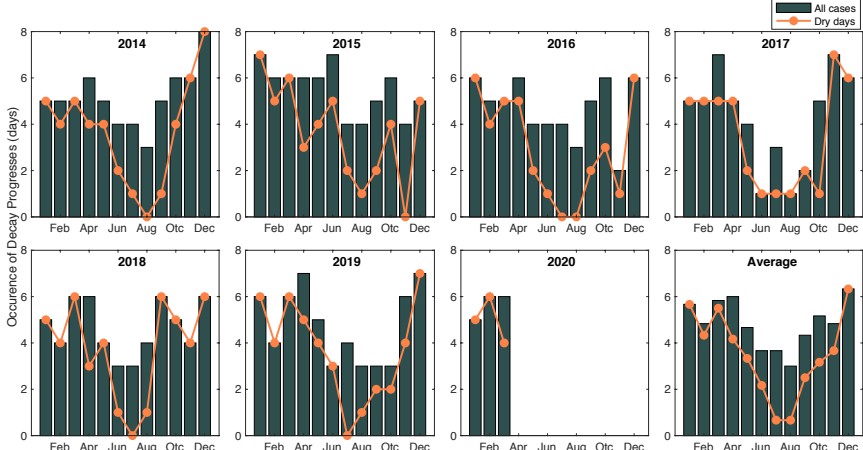


Figure 4. Monthly cumulative occurrence of the decay processes of pollution episodes. The orange
curve indicates the decay process occurrences on dry days. In total, 365 decay processes are
identified from January 2014 to March 2020, and 97 of them are associated with precipitation levels
greater than 10 mm/day.



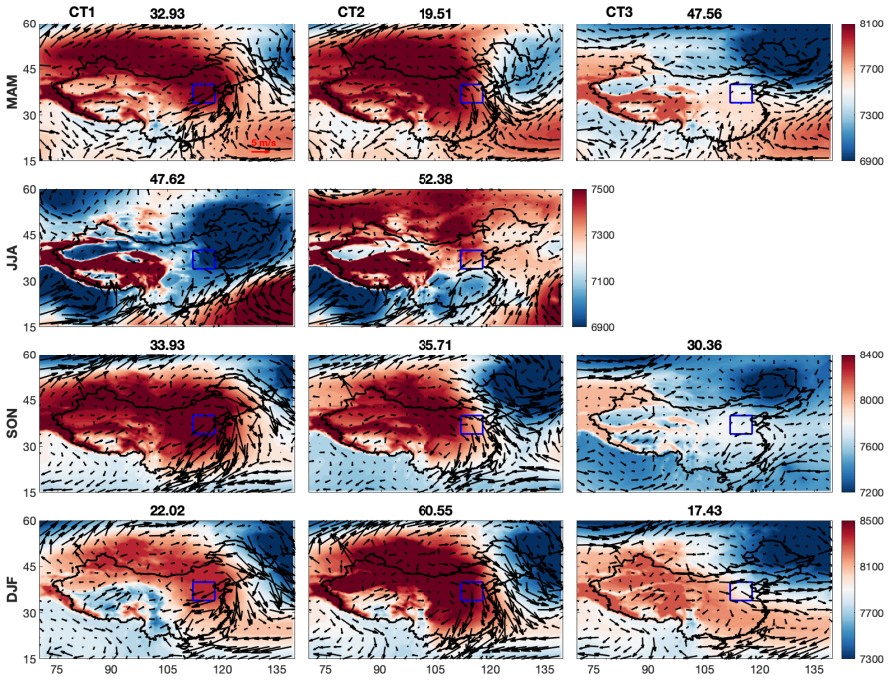

Figure 5. Distribution of the geopotential height (shaded, units: $m^2/s^2$) and wind fields at 925 hPa for each circulation type. The number over each subplot indicates the occurrence frequency of the specific circulation type. The solid blue box is the location of the domain region covering the 28 pollution channel cities.





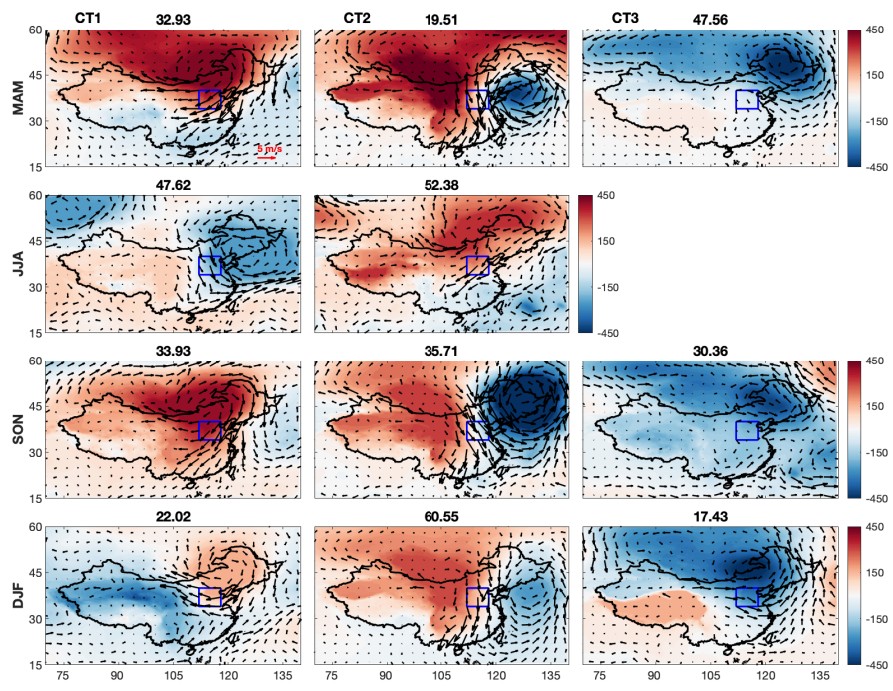


Figure 6. Distribution of the geopotential height anomalies (shaded, unit: m²/s²) and wind field
anomalies at 925 hPa for each circulation type.

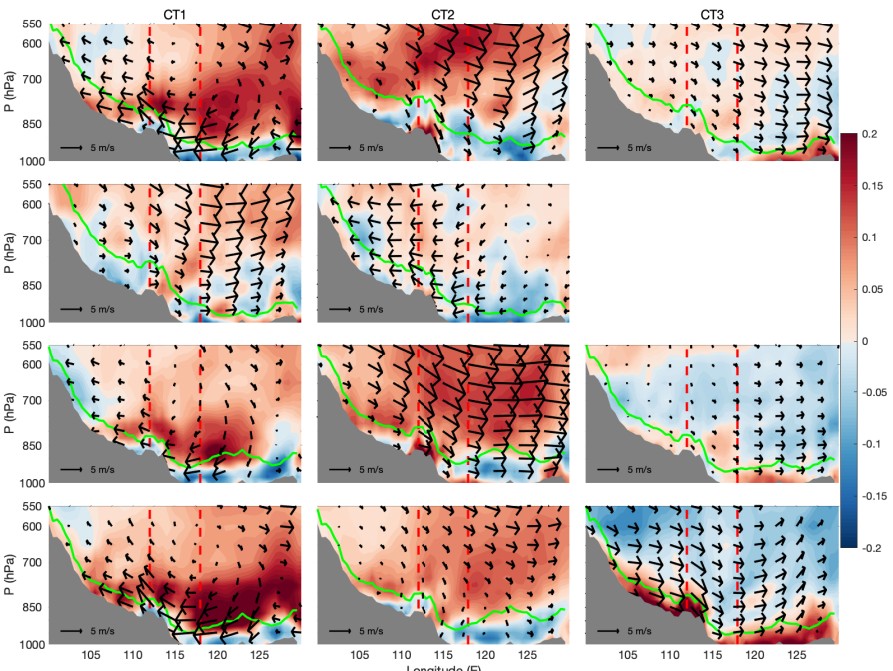


Figure 7. Zonal averaged profile of the distribution of vertical wind shear anomalies in the domain
region (shaded, units: m/(s.100 m)) and the vertical and zonal circulation anomalies. The green line
indicates the average location of the top of the boundary layer. Zonal wind shear, circulation and
boundary layer height are the average values between 34-40° N. The two dashed lines are the eastern
and western boundaries of the domain (112 to118° E). The grey region indicates the average altitude
between 34-40° N.



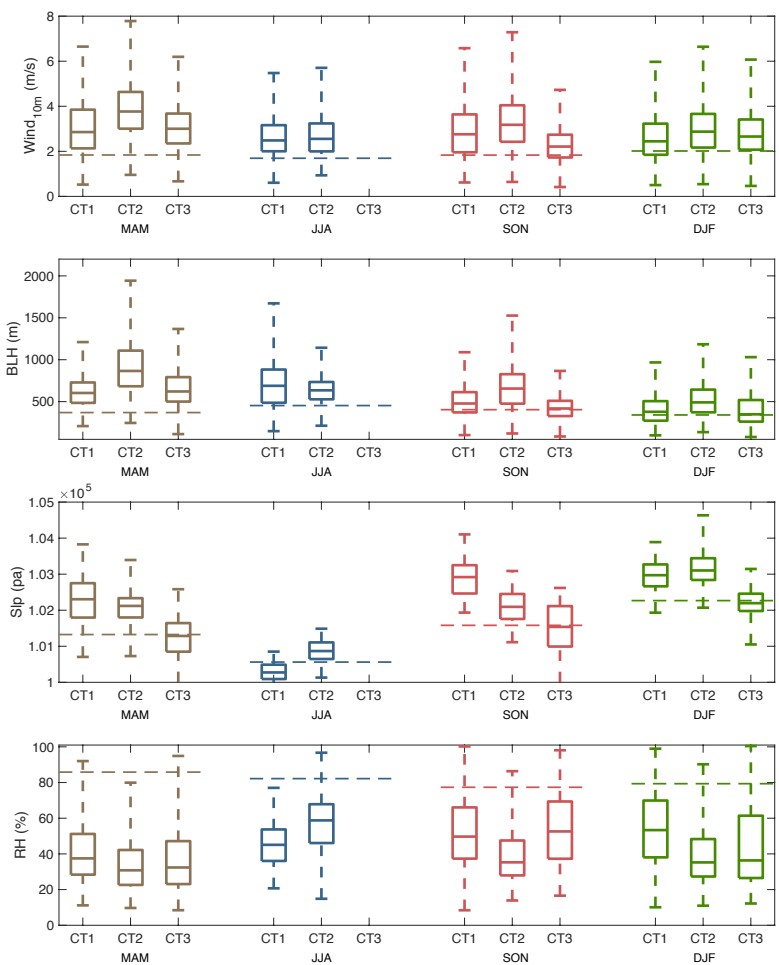


Figure 8. Boxplot of surface wind speed, boundary layer height (BLH), sea level pressure (slp) and

relative humidity (RH) for each circulation type. The dashed line indicates the seasonal mean of the

specific variables.



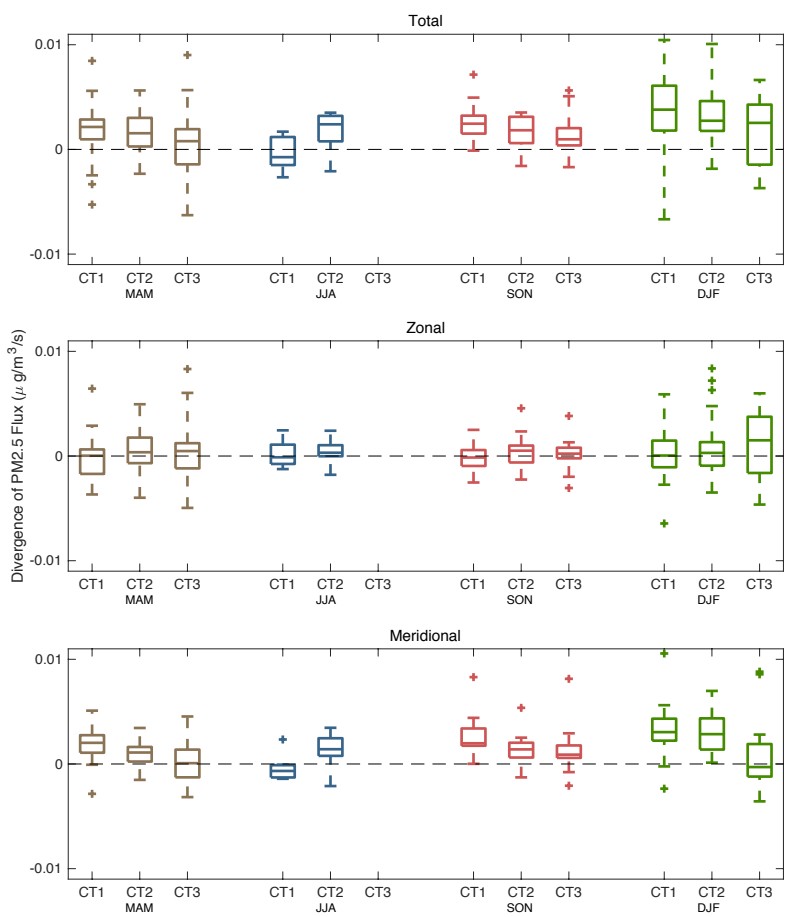


Figure 9. Boxplot of the divergence of PM2.5 flux over the region of 34-40° N and 112-118° E. The

daily divergence is calculated based on the Eq. (1). Zonal and meridional components are the first

and second terms of the formula.





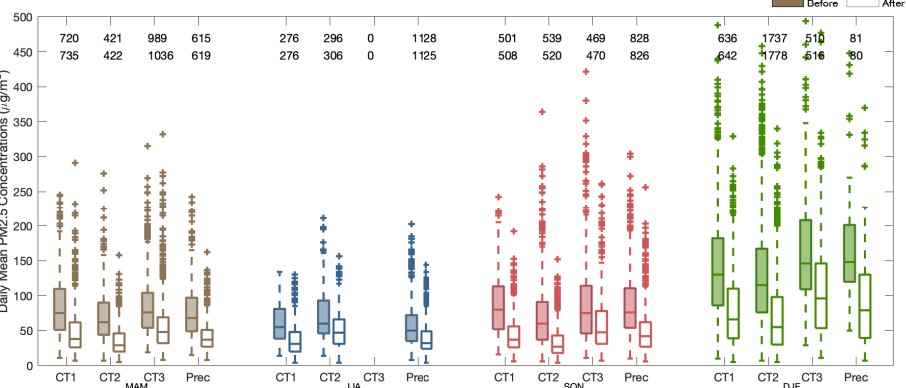


Figure 10. Distribution of the daily mean PM2.5 concentrations before and after the occurrence of decay processes of pollution episodes in the 28 pollution channel cities. The hollow box indicates the concentration on the decay phase day, and the solid box is the value on the previous day. The relative differences in the PM2.5 concentrations after the occurrence of decay process are summarized in Table 2. The number at the top of each box indicates the sample size used for the boxplot. The number in the first line is the sample size of the "before" case; and the second line is for the "after" case.






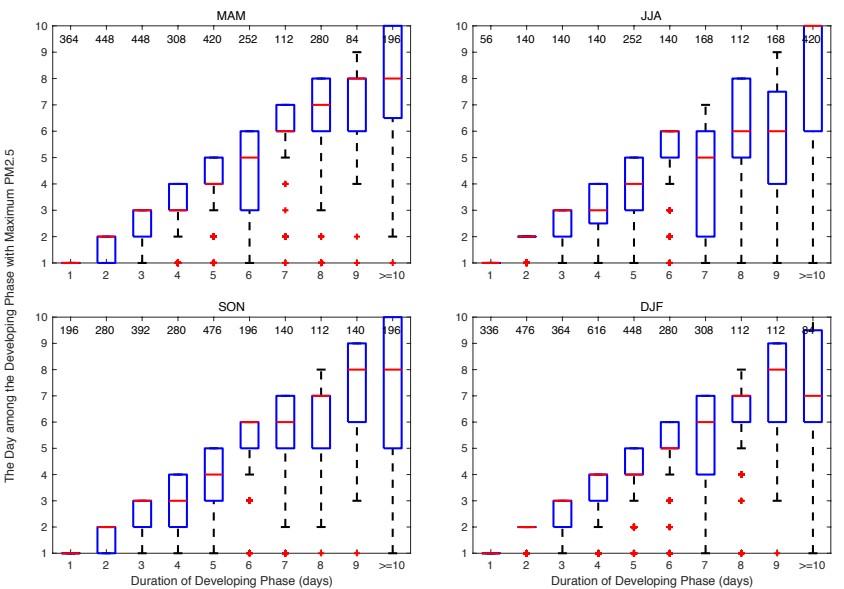


Figure 11. The day of the maximum PM2.5 concentration during each pollution episode varies with
the duration of the developing phase.



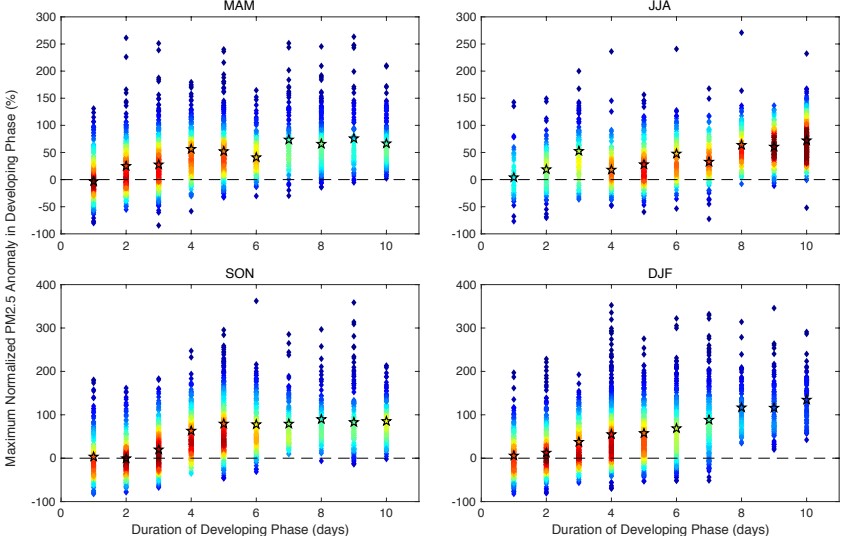


Figure 12. The density plot of the maximum PM2.5 concentration according to the duration of the

developing phase of pollution episodes. Daily PM2.5 concentrations are normalized by their

monthly mean value to exclude the effects of seasonal and interannual variations in air quality. A

warmer color indicates a higher density of scatter. Pentagrams mark the average maximum PM2.5

concentration for the specific duration period.

787



788

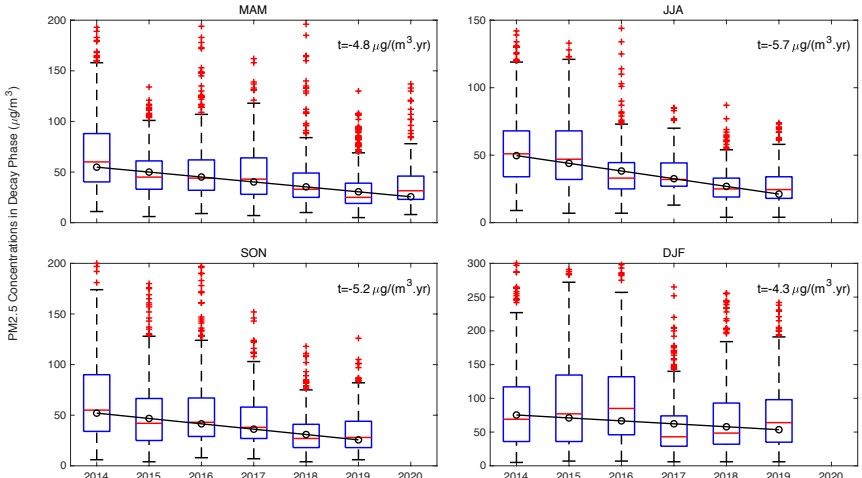

789

Figure 13. Variations in the average PM2.5 concentration on all the decay phase days from 2014 to

2020. The black hollow circles indicate the mean PM2.5 concentration in each year. The black line

is the fitting line based on the mean value. The number in the subplot is the linear trend of the fitting

line.

794