# Peer review of "Dominant synoptic patterns associated with the decay process of PM2.5 pollution episodes around Beijing"

_Atmospheric Chemistry and Physics, 2020_

## Referee Comment (RC1) · Anonymous Referee #1 · 19 Sep 2020

1. The introduction lacks of progresses about research issues.

2. The colors in Figure 3 make the readers confused, need to be modified.

3. Line 149, "which is known as a sawtooth cycle", the adjustment interval of synoptic circulations is related to the period of Rossby waves, which is about 1 week.

4. Line 175, why not to choose other weather typing approaches, such as Lamb-Jenkinson method. It's have no associations with sample data.

5. Line 270, strong vertical shear? The top of PBL was located between positive and negative wind shear in most cases where the shear is about zero. Have you compared

with pollution situations? This may be caused by the air with different properties within or above the PBL.

6. Line 412, horizontal wind shear, do you mean vertical shear of horizontal winds?

7. About the primary conclusions in paper, we have already known that meteorological conditions, such as strong winds and low relative humidity, are favorable for removing pollutants. However, in CT1, pollutants within PBL diffused upward, while flows presented to be sinking motion in CT2 and CT3 in contrast. What is the leading mechanism for removing pollutants in different patterns?

8. How much is the contribution of horizontal advection and vertical diffusion respectively for removing pollution in different types and seasons? And mainly through which layer do pollutants diffuse downstream?

9. The whole paper needs to be more standard and concise. Some conclusions are moved to the discussion section. The removing mechanisms in different synoptic patterns need to be more clarified.

---

## Referee Comment (RC2) · Anonymous Referee #2 · 26 Oct 2020

This manuscript studied the meteorological factors that contribute to the decay process of the PM2.5 pollution episodes in Beijing and its surrounding regions. They identified three dominant circulation types that favor the decay of high concentrations of pollutants, using the T-mode PCA analysis of geopotential height and horizontal winds during the selected PM2.5 decay days. The topic aligns well with the scope of the journal. The manuscript is well-written with minor corrections of some sentences as indicated below. The method is robust and conclusions are well supported by data and results. My only suggestion is adding discussions about the importance of the study in the context of air quality as well as caveats at the end of this manuscript.

[Figure]

1. Lines 28–30, "the same three circulation types (CTs)..." does not make sense, cause only two CTs were mentioned before.

2. In Abstract, it's better to define CT1, CT2, and CT3 first and then discuss their impacts on PM2.5 decay processes.

3. Line 130, it should be "in a specific region,"

4. Line 208, delete "of".

5. How a dry day is defined? Is it defined for each grid cell or for the entire study domain? Is it defined as a day with zero precipitation or with precipitation less than a threshold? What precipitation data were used?

6. Line 211, should be "a specific year".

7. Figure 8, the four variables in each circulation type and each season should be tested to see if they are statistically different from the corresponding seasonal means. The variables that past the significant test should be highlighted in the figure and described in the text.

8. Figure 13, the method to estimate the linear trend should be mentioned and corresponding p values or uncertainties of these trends should be included.

9. Figure 5 gives similar information to Figure 6, and may be moved to supplementary document.

10. Figure 1 can be modified to add topography information as shadings, since topography is also an important factor that influences the dilution of the pollutants.

11. Figure 2 can be improved by showing the mean across the 28 cities with shadings indicating the range of PM2.5. The current figure is a little noisy to observe the sharp decay process.

12. It is interesting to show the corresponding time series (i.e., principle components)

of each circulation type in Figure 5 and to check if there are any temporal trends. If there are trends, then the decreasing trends in Figure 13 can be partially attributed to circulation changes besides emission changes.
* * *

---

## Author Comment (AC2) · 18 Dec 2020

The comment was uploaded in the form of a supplement:
https://acp.copernicus.org/preprints/acp-2020-912/acp-2020-912-AC2-supplement.pdf

---

## Author Response (AR2)

**Reviewer #1**

**Comment 1:** The introduction lacks of progresses about research issues.

**Response 1:** We overviewed some more research progresses in Lines 90-93: *Circulation of a strong Siberian High to the north and cold anomalies in the low-level troposphere with strong East Asian Trough is found to be favorable for the clear winter in Beijing and surrounding region (Pei and Yan, 2018).*

And Lines 107-117: *However, compared to the developing phase, which typically features a smooth increase in air pollutant concentrations due to the regional transport, local accumulation and secondary formation, the decay phase of each pollution episode shows a sharp decrease in PM2.5 concentrations, often in a few hours. Pollutants on hazy days show mass concentration 2-3 times higher than that on clear days (Li et al., 2010). The abrupt decrease in PM2.5 concentrations is purely meteorological in origin and is controlled by the passage of synoptic systems, especially cold fronts, which terminate a severe air pollution episode in the BTH region by strong winds (Zhu et al., 2016;Jia et al., 2008;Ji et al., 2012;Xin et al., 2012). Many studies took the smooth increase period of PM2.5 concentrations and abrupt decrease stage following it as a complete air pollution episode, and investigate its development mechanism (Tang et al., 2016b; Zhang et al., 2018b;Sun et al., 2014;Zheng et al., 2015).*

Reference:

Pei, L., and Yan, Z.: Diminishing clear winter skies in Beijing towards a possible future, Environ. Res. Lett., 13, 124029, 2018.

Li, W., Shao, L., and Buseck, P.: Haze types in Beijing and the influence of agricultural biomass burning, Atmos. Chem. Phys., 10, 2010.

Zhu, X., Tang, G., Hu, B., Wang, L., Xin, J., Zhang, J., Liu, Z., Münkel, C., and Wang, Y.: Regional pollution and its formation mechanism over North China Plain: A case study with ceilometer observations and model simulations, J. Geophys.Res. Atmos., 121, 14,574-514,588, 2016.

Jia, Y., Rahn, K. A., He, K., Wen, T., and Wang, Y.: A novel technique for quantifying the regional component of urban aerosol solely from its sawtooth cycles, J. Geophys.Res. Atmos., 113, D21309, 2008.

Ji, D., Wang, Y., Wang, L., Chen, L., Hu, B., Tang, G., Xin, J., Song, T., Wen, T., and Sun, Y.: Analysis of heavy pollution episodes in selected cities of northern China, Atmos. Environ., 50, 338-348, 2012.

Xin, J., Gong, C., Wang, S., and Wang, Y.: Aerosol direct radiative forcing in desert and semi-desert regions of northwestern China, Atmos. Res., 171, 56-65, 2016.

Tang, L., Yu, H., Ding, A., Zhang, Y., Qin, W., Wang, Z., Chen, W., Hua, Y., and Yang, X.: Regional contribution to PM1 pollution during winter haze in Yangtze River Delta, China, Sci. Total Environ., 541, 161-166, 2016b.

Zhang, X., Zhong, J., Wang, J., Wang, Y., and Liu, Y.: The interdecadal worsening of weather conditions affecting aerosol pollution in the Beijing area in relation to climate warming, Atmos. Chem. Phys., 18, 5991-5999, 2018b.

Sun, Y., Jiang, Q., Wang, Z., Fu, P., Li, J., Yang, T., and Yin, Y.: Investigation of the sources and evolution processes of severe haze pollution in Beijing in January 2013, J. Geophys.Res. Atmos., 119, 4380-4398, 2014.

Zheng, G., Duan, F., Su, H., Ma, Y., Cheng, Y., Zheng, B., Zhang, Q., Huang, T., Kimoto, T., and Chang, D.: Exploring the severe winter haze in Beijing: the impact of synoptic weather, regional transport and heterogeneous reactions, Atmos. Chem. Phys., 15, 2969, 2015.

**Comment 2:** The colors in Figure 3 make the readers confused, need to be modified.

**Response 2:** The probability density function of day-to-day difference of PM2.5 in Figure 3 has been modified in the revised version.

[Figure]

Figure 3. Probability distribution of the relative day-to-day difference of PM2.5 concentrations. The relative difference is based on the PM2.5 concentration on the previous day. The distributions in spring and autumn are combined in the upper panel, and cases in winter and summer are shown at the bottom.

**Comment 3:** Line 149, "which is known as a sawtooth cycle", the adjustment interval of synoptic circulations is related to the period of Rossby waves, which is about 1 week.

**Response 3:** Yes, Rossby wave is the dominating wave system in the mid-latitude region, which has a cycle of about one week. The average interval durations of two decay processes are ranged in 5.36 to 7.45 days in different seasons, according with the cycle length of Rossby wave. Therefore, we added some description in Lines 370-372 to link the decay phase duration and Rossby wave cycle: *As the main wave system affecting the synoptic circulation in mid-latitude region, the Rossby wave has about one-week cycle length, which dominates the average duration of two adjacent decay*

*phase.*

And in Lines 410-412: *The intervals between two continuous decay processes are 5.53, 7.45, 5.86 and 5.36 days from spring to winter, respectively, which may be controlled by the cycle length of Rossby waves in the mid-latitude region.*

**Comment 4:** Line 175, why not to choose other weather typing approaches, such as Lamb-Jenkinson method. It's has no associations with sample data.

**Response 4:** Thanks for your suggestion. We tried the method of Lamb-Jenkinson method in the revised version. The circulation patterns based on the new classification method are similar to that of the principal component analysis (PCA) method as shown in the Supplementary Information (text in SI and Fig. S2-S3), which confirmed the robust of the classification results of PCA method.

We add some description about the Lamb- Jenkinson method in SI: *ERA5 daily mean sea level pressure in Jan. 2014 to Mar. 2020 are used to classify the synoptic circulation types based on Lamb-Jenkinson method. Circulation types are classified into 26 types including eight directional types (northerly, N; northeasterly, NE; easterly, E; southeasterly, SE; southerly, S; southwesterly, SW; westerly, W; and northwesterly, NW), two vorticity types (cyclonic, C; anticyclonic, A) and sixteen hybrid types (CN, CNE, CE, CSE, CS, CSW, CNW, AN, ANE, AE, ASE, AS, ASW, AW and ANW). Figure S2 shows the seasonal frequency of the 26 CTs during Jan. 2014 to Mar. 2020. The frequencies of the two vorticity and eight directional types were much higher than those of other sixteen hybrid CTs. The top four highest frequency for the specific season are highlighted in Fig. S2. Fig. S3 shows the distribution of atmospheric circulation at 925 hPa of the top four highest frequency CTs. Circulation characteristics in Figure S1 has the similar pattern with those in Figure S3, which indicates the robust of the two circulation classification methods.*

[Figure]

*Figure S2. Occurrence frequency of 26 kinds of circulation types based on Lamb-Jenkinson circulation classification method during Jan. 2014 to Mar. 2020. Red dots indicate the top four highest frequency.*

[Figure]

*Figure S3. Distribution of the geopotential height (shade, units: $m^2/s^2$) and wind fields at 925 hPa for the top four highest frequency CTs based on Lamb-Jenkinson classification methods. The title of*

*each subplot indicates the specific CTs and the corresponding frequency (%) in each season.*

We also add some description in Lines 193-202 of the main text as: *The Lamb-Jenkinson-Collison type classification (JCT) is also a widely adopted method to identify synoptic circulation pattern by describing the location of cyclonic/anticyclonic centers and the direction of the geostrophic flow (Li et al., 2020;Fan et al., 2015;Jiang et al., 2020;Chen, 2000;Jenkinson and Collison, 1977). In order to verify the robust of circulation classification results of PCA method, JCT method is also involved based on daily mean gridded sea level pressure at 16 points centered by 37° N and 117° E as shown in SI. According to Fig. S1 and Fig. S3, it shows almost the similar circulation pattern of PCA and JCT method, indicating the consistence of two classification methods. Because JCT method is specialized on classifying daily mean sea level pressure patterns, which will ignore the thresholds of some other meteorological variables to some extent. Therefore, we only focus on the results of PCA hereafter.*

Reference:

Li, M., Wang, L., Liu, J., Gao, W., Song, T., Sun, Y., Li, L., Li, X., Wang, Y., and Liu, L.: Exploring the regional pollution characteristics and meteorological formation mechanism of PM2. 5 in North China during 2013–2017, Environ. Int., 134, 105283, 2020.

Fan, L., Yan, Z., Chen, D., and Fu, C.: Comparison between two statistical downscaling methods for summer daily rainfall in Chongqing, China, Int. J. Climatol., 35, 3781-3797, 2015.

Jiang, Y., Xin, J., Wang, Y., Tang, G., Zhao, Y., Jia, D., Zhao, D., Wang, M., Dai, L., and Wang, L.: The dynamic-thermal structures of the planetary boundary layer dominated by synoptic circulations and the regular effect on air pollution in Beijing, Atmos. Chem. Phys. Discuss., 1-21, 2020.

Chen, D.: A monthly circulation climatology for Sweden and its application to a winter temperature case study, Int. J. Climatol., 20, 1067-1076, 2000.

Jenkinson, A., and Collison, F.: An initial climatology of gales over the North Sea, Synoptic climatology branch memorandum, 62, 18, 1977.

Philipp, A., Beck, C., Esteban, P., Kreienkamp, F., Krennert, T., Lochbihler, K., Lykoudis, S. P., Pianko-Kluczynska, K., Post, P., and Alvarez10, D. R.: cost733class-1.2 User guide, Augsburg, Germany, 10-21, 2014.

**Comment 5:** Line 270, strong vertical shear? The top of PBL was located between positive and negative wind shear in most cases where the shear is about zero. Have you compared with pollution situations? This may be caused by the air with different properties within or above the PBL.

**Response 5:** Yes, in CT1 condition, the top of the boundary layer located at the transition zone from negative wind shear anomaly to positive anomaly. While, due to westward climbing of the

prevailing easterly wind, low-level air pollutants are taking out of the boundary layer. As long as the air pollutants are brought to the free atmosphere, they will be well blended quickly due to the upper-level strong wind shear.

According to your suggestion, 24 h backward trajectories of Beijing are conducted based on NOAA HYSPLIT Trajectory model using NECP reanalysis dataset. Fig. S6 shows the surface (10 m) and free atmosphere (2000 m above ground level) 24 h backward trajectories of each decay phase day. We can find that the air mass within and above the boundary layer almost come from the same direction in a specific circulation type, which indicates the consistence air mass properties from the surface to the free atmosphere. In addition, most of the backward trajectories come from west and northwest of Beijing, which brings cold and dry air mass and benefits for the decay of pollution episodes. We add the description in Lines 302-303: *According to the distribution of 24 h backward trajectories of Beijing in Fig. S6, the northwesterly cold and dry air mass are taking to the domain, benefiting for the decay of local pollution episodes.*

[Figure]

Figure S6. 24 h backward trajectories of Beijing at 10 m and 2000 m (above ground level) on all the decay phase days based on the NOAA HYSPLIT Trajectory model.

**Comment 6:** Line 412, horizontal wind shear, do you mean vertical shear of horizontal winds?

**Response 6:** Yes, it should be vertical shear of horizontal winds. We have revised it in the new version.

**Comment 7:** About the primary conclusions in paper, we have already known that meteorological conditions, such as strong winds and low relative humidity, are favorable for removing pollutants. However, in CT1, pollutants within PBL diffused upward, while flows presented to be sinking motion in CT2 and CT3 in contrast. What is the leading mechanism for removing pollutants in different patterns?

**Response 7:** The prevailing wind direction is easterly wind in CT1, which will climb westward due to the western mountain region and in turn brings the surface air pollutants out of the boundary layer. While, in CT2 and CT3 conditions, the air mass crossing the mountain will motivate a downward motion, which is known as the downwash airflow due to blocking of mountainous terrain. In CT2, it is the strongest positive wind anomaly, positive boundary layer heigh anomaly and negative relative humidity that leads to the quickly decay of air pollution, which is the commonly believed synoptic circulation breaking off pollution episodes over northern China. The cleaning performance of CT2 is the strongest among the three kinds of circulation types. CT3 also has positive wind speed anomaly and negative relative humidity anomaly, but the magnitudes of anomalies are not as significant as those of CT2. Stronger than normal vertical mixing within the boundary layer may contribute to the decrease in air pollutant concentrations. Although PM2.5 concentration will decrease by 27~29% after the passage of CT3, the removal efficiency of CT3 is the weakest among the three CTs, which can be attributed to the moderate favorable horizontal diffusion conditions. We clarified the cleaning mechanisms in the Conclusion section in Lines 413-427: *All the CTs are common in positive wind speed anomaly, negative relative humidity anomaly and effective outflow of PM2.5 from the domain. Although the magnitude and significance of the anomalies are different in the specific CT, all the above variables indicate favorable atmospheric diffusion conditions, which is benefit for the decay of pollution episodes. There are also some prominent features for each CT. In CT1, the most significant horizontal outflow of air pollutants combining with the upward transport of airflow to the free atmosphere are the two extra drivers for the decay processes. The removal efficiency of CT1 is 35-40%, which is moderate among the three CTs. In terms of CT2, it is the most frequent CT in autumn and winter. The circulation with the heaviest wind speed from the northwest, the highest BLH, lowest relative humidity jointly results in the quickly decrease in PM2.5 concentration in a few hours, which is the commonly accepted circulation feature to terminate the severe pollution episodes. Due to the significantly favorable meteorological conditions, CT2 has the strongest cleaning abilities of 41-45% in different seasons. For CT3, the synergy effects of enhanced vertical mixing within the boundary layer and moderate beneficial background of wind speed, relative humidity and horizontal divergence of PM2.5 are the main cleaning mechanism of CT3 condition.*

And add some discussions in Lines 440-444: *PM2.5 concentrations sharply decrease after the*

*passage of CT2, but it shows a relatively weak drop in air pollutant concentrations when CT3 occurs, which can be attributed to its moderate strength of anomalies circulation pattern. Therefore, the scavenging effects of each CT should also be taken into account when predicting the air quality based on synoptic circulation variation.*

**Comment 8:** How much is the contribution of horizontal advection and vertical diffusion respectively for removing pollution in different types and seasons? And mainly through which layer do pollutants diffuse downstream?

**Response 8:** The net PM2.5 flux of horizontal and vertical direction is an ideal metric to evaluate the outflow or inflow of air pollutants from a domain. High temporal (daily at least) and spatial resolution (grid scale at least) PM2.5 profile and 3-D wind fields are needed to measure the contribution of vertical diffusion. However, the grid scale daily PM2.5 profiles are not available currently. Satellite data, e.g., CALIPSO Level 2 aerosol profiles, could provide aerosol profiles with 5 km horizontal resolution, but its limit temporal resolution (with a repeat cycle of 16-day) does not meet the requirement. We added the discussion about this in Lines 435-440: *Due to the limitation of dataset about PM2.5 vertical distribution, only the horizontal divergence of PM2.5 flux is used in this study. Although it shows positive divergence for all the CTs, indicating the remarkable contribution of the net outflow of air pollutants at the surface to the quickly decrease in PM2.5 concentrations, the effects of horizontal PM2.5 flux above the surface or the vertical diffusion cannot be neglected, which may have great contribution in a specific event, and need to be further studies.*

In addition, the horizontal divergence of PM2.5 flux is further refined to four directions in Fig. S4, which shows more detailed information of flux from each side. For CT1, the horizontal PM2.5 flux divergence is the most positive, with significant outflow of air pollutants from in the southern edge of the domain. The magnitude of inflow from eastern side is at the same level as the outflow from western edge, leading to the insignificant zonal divergence. For CT2, significant positive divergence in the eastern and southern edges contribute to the net outflow of air pollutants. In terms of CT3, the zonal divergence of the PM2.5 flux dominates the net positive divergence of the whole region, rather than the meridional component as the other two circulation patterns.

[Figure]

Figure S4. Boxplot of the divergence of PM2.5 flux from the four sides of the region of 34-40° N and 112-118° E. Positive divergence indicates outflow of PM2.5 from the specific direction; negative divergence indicates inflow of PM2.5 from the domain. * in the x axis marks the divergence in a specific CT is significantly different with zero based on two-tail student-t test at a significant level of 0.01.

**Comment 9:** The whole paper needs to be more standard and concise. Some conclusions are moved to the discussion section. The removing mechanisms in different synoptic patterns need to be more clarified.

**Response 9:** Thanks for your suggestion. We reorganized the abstract and conclusion sections to clarify the removing mechanism of each circulation type.

Abstract section in Lines 35-39: *CT2 is the most frequent CT in autumn and winter, with the highest wind speed from the northwest, the highest boundary layer height (BLH), and lowest relative humidity among the three CTs, all of which are favorable for the reduction of PM2.5 concentrations. In CT3, strong vertical wind shear within the boundary layer enhances the mixing of surface air*

*pollutants, which is the extra cleaning mechanism besides dry and clean air mass inflow.*

Conclusion section in Lines 413-444: *All the CTs are common in positive wind speed anomaly, negative relative humidity anomaly and effective outflow of PM2.5 from the domain. Although the magnitude and significance of the anomalies are different in the specific CT, all the above variables indicate favorable atmospheric diffusion conditions, which is benefit for the decay of pollution episodes. There are also some prominent features for each CT. In CT1, the most significant horizontal outflow of air pollutants combining with the upward transport of airflow to the free atmosphere are the two extra drivers for the decay processes. The removal efficiency of CT1 is 35-40%, which is moderate among the three CTs. In terms of CT2, it is the most frequent CT in autumn and winter. The circulation with the heaviest wind speed from the northwest, the highest BLH, lowest relative humidity jointly results in the quickly decrease in PM2.5 concentration in a few hours, which is the commonly accepted circulation feature to terminate the severe pollution episodes. Due to the significantly favorable meteorological conditions, CT2 has the strongest cleaning abilities of 41-45% in different seasons. For CT3, the synergy effects of enhanced vertical mixing within the boundary layer and moderate beneficial background of wind speed, relative humidity and horizontal divergence of PM2.5 are the main cleaning mechanism of CT3 condition. After the passage of CT3, 26~29% of local air pollutants are typically removed. The two dry-day circulation patterns in summer are similar to the synoptic patterns of CT1 and CT3 in the other three seasons. A dry air mass with a positive BLH anomaly and the effective horizontal outflow of air pollutants are the main reasons for the abrupt decay phases in summer. The average PM2.5 concentrations on decay process days show a significant decreasing trend from 2014 to 2020, which indicates the success of emission mitigation efforts. Emission reductions have led to a 4.3~5.7 $\mu g/(m^3.yr)$ decrease in PM2.5 concentrations in the 28 pollution channel cities.*

*Due to the limitation of dataset about PM2.5 vertical distribution, only the horizontal divergence of PM2.5 flux is used in this study. Although it shows positive divergence for all of the CTs, indicating the remarkable contribution of the net outflow of air pollutants at the surface to the quickly decrease in PM2.5 concentrations, the effects of horizontal PM2.5 flux above the surface or the vertical diffusion cannot be neglected, which may have great contribution in a specific event, and need to be further studies. PM2.5 concentrations sharply decrease after the passage of CT2, but it shows a relatively weak drop in air pollutant concentrations when CT3 occurs, which can be attributed to its moderate strength of anomalies circulation pattern. Therefore, the scavenging effects of each CT should also be taken into account when predicting the air quality based on synoptic circulation variation.*

**Reviewer #2**

**Comment 1:** Lines 28–30, "the same three circulation types (CTs)..." does not make sense, cause only two CTs were mentioned before.

**Response 1:** We revised the sentence as in Lines 29-30 "*Two dominant circulation patterns are identified in summer. All the other three seasons have three circulation types (CTs), respectively. The three CTs in spring show the same patterns with those in autumn and winter*".

**Comment 2:** In Abstract, it's better to define CT1, CT2, and CT3 first and then discuss their impacts on PM2.5 decay processes.

**Response 2:** Thanks for your suggestion. We reorganized the removing mechanisms of each CT in the abstract section in Lines 30-41: *The circulation patterns beneficial to the decay processes all exhibit a higher-than-normal surface wind speed, a negative relative humidity anomaly and net outflow of PM2.5 from the domain. In addition, CT1 in spring, autumn and winter is controlled by northeasterly wind and features the most significant horizontal net-outflow of air pollutants and effective upward spread of air pollutants to the free atmosphere. CT2 is the most frequent CT in autumn and winter, with the highest wind speed from the northwest, the highest boundary layer height (BLH), and lowest relative humidity among the three CTs, all of which are favorable for the reduction of PM2.5 concentrations. In CT3, strong vertical wind shear within the boundary layer enhances the mixing of surface air pollutants, which is the extra cleaning mechanism besides dry and clean air mass inflow. PM2.5 concentrations show significant decreases of more than 37%, 41% and 27% after the passage of CT1, CT2 and CT3, respectively.*

**Comment 3:** Line 130, it should be "in a specific region,"

**Response 3:** Revised as suggested.

**Comment 4:** Line 208, delete "of".

**Response 4:** Revised as suggested.

**Comment 5:** How a dry day is defined? Is it defined for each grid cell or for the entire study domain? Is it defined as a day with zero precipitation or with precipitation less than a threshold? What precipitation data were used?

**Response 5:** ERA5 hourly total precipitation dataset with a resolution of 0.5°*0.5° is used in this study. The domain region covering the 28 cities is 36°-42° N and 113°-117.5° E. If the daily mean accumulate precipitation amount is larger than 1 mm for all the grid cells in the domain, the day is defined as a rainy day with effective wet deposition. We added the defined of dry days in the text in Line 135-136: *Daily accumulated precipitation amount is the total amount of 24-hour values.*

And in Lines 224-228: *If the daily mean accumulated precipitation amount is more than 1 mm for all the grid cells in the region of 36°-42° N and 113°-117.5° E (covering the 28 cities), the specific day is defined as a rainy day with effective wet deposition. 97 of the 365 decay phases are defined as rainy days, in which case the abrupt decrease in ambient PM2.5 concentrations are assumed to be related to wet deposition.*

**Comment 6:** Line 211, should be "a specific year".

**Response 6:** Revised as suggested.

**Comment 7:** Figure 8, the four variables in each circulation type and each season should be tested to see if they are statistically different from the corresponding seasonal means. The variables that past the significant test should be highlighted in the figure and described in the text.

**Response 7:** The mean values for all the four meteorological variables in each circulation in original Fig. 8 (Fig. 7 in the new version) are significantly different with their seasonal mean based on two-tail student-t test at a significant level of 0.01. The result of significant test was added in the figure caption of Fig. 7. In addition, the student-t test is also conducted to the divergence distribution of PM2.5 flux in Fig. 8 and Fig. S2. * is used to highlight the circulation with significant positive or negative divergence.

**Comment 8:** Figure 13, the method to estimate the linear trend should be mentioned and corresponding p values or uncertainties of these trends should be included.

**Response 8:** Least squares regression is used to estimate the linear trend of the monthly median PM2.5 variations. R-square and p-value for each of the regression model was involved in Fig. 12 (original Fig. 13), which shows significant decrease in PM2.5 concentrations in spring, summer, autumn during 2014 to 2019. The corresponding description was added in the figure caption.

[Figure]

Figure 12. Variations in the average PM2.5 concentration on all the decay phase days from 2014 to 2020. The black hollow circles indicate the mean PM2.5 concentration in each year. The black line is the fitting line based on the montly median value. The number in the subplot is the linear trend (t), R-square and p-value of least squares regression model. ** after linear trend indicates the linear regression model is significant with a p-value<0.01.

**Comment 9:** Figure 5 gives similar information to Figure 6, and may be moved to supplementary document.

**Response 9:** Fig. 5 has been moved to the supplementary information in the revised version. Moreover, the distribution of geopotential height anomalies at 500 hPa was also involved in the supplementary.

**Comment 10:** Figure 1 can be modified to add topography information as shadings, since topography is also an important factor that influences the dilution of the pollutants.

**Response 10:** Thanks for your suggestion. The terrain distribution based on the Global Digital Elevation Model was added in Fig. 1.

[Figure]

Figure 1. Distribution of annual mean PM2.5 concentrations in the 28 cities by altitude. The PM2.5 concentration is the annual mean value from 2014 to 2019 (units: μg/m³). The elevation over the domain was obtained from Global Digital Elevation Model with a resolution of 0.5°*0.5°.

**Comment 11:** Figure 2 can be improved by showing the mean across the 28 cities with shadings indicating the range of PM2.5. The current figure is a little noisy to observe the sharp decay process.

**Response 11:** Thanks for your suggestion. We revised Fig. 2 with PM2.5 concentrations at the 28 cities displaying by their range.

[Figure]

Figure 2. Time series of daily mean PM2.5 concentrations in the 28 pollution channel cities from

January to March 2019 (units: μg/m³).

**Comment 12:** It is interesting to show the corresponding time series (i.e., principle components) of each circulation type in Figure 5 and to check if there are any temporal trends. If there are trends, then the decreasing trends in Figure 13 can be partially attributed to circulation changes besides emission changes.

**Response 12:** The time series of each circulation type (CT) frequency can be obtained by year or season as shown in Fig. R1, which has no obvious correspondence with PM2.5 variations in Fig. 12 (original Fig. 13). The interannual variation of seasonal mean PM2.5 concentrations may be closely related to the change of seasonal occurrence frequency of a specific CT or their accumulated frequency. In addition, the average interval between two decay progresses may also affect the final seasonal mean PM2.5 concentrations. We have analyzed the combined effects of CT frequency and their interval on the interannual variation of PM2.5 concentrations in our previous work (doi.org/10.5194/acp-20-7667-2020). However, Fig. 12 shows the interannual variation of PM2.5 concentrations only on the decay phase days instead of the seasonal mean value. If we suppose it has the same scavenging ability for all the decay phase CTs, i.e., the effects of meteorological conditions to air quality remain the same; PM2.5 after the decay phase would indicate the background air pollutant concentration of ambient environment. Therefore, the long-term variation of the background concentrations can represent the change of emission to a certain extent.

[Figure]

Figure R1. Interannual variation of seasonal occurrence frequency for each CT (left axis) and accumulated frequency for all the three CTs (right axis).